# Psychosocial resources predict frequent pain differently for men and women: A prospective cohort study

Anke Samulowitz[1]*, Inger Haukenes[2,3], Anna Grimby-Ekman[1], Stefan Bergman[1,4], Gunnel Hensing[1]

1 School of Public Health and Community Medicine, Institute of Medicine, The Sahlgrenska Academy, University of Gothenburg, Gothenburg, Sweden, 2 Department of Global Public Health and Primary Care, University of Bergen, Bergen, Norway, 3 Research Unit for General Practice, NORCE Norwegian Research Centre, Bergen, Norway, 4 Spenshult Research and Development Centre, Halmstad, Sweden

* anke.samulowitz@gu.se

## Abstract

### Introduction

Psychosocial resources, psychological and social factors like self-efficacy and social support have been suggested as important assets for individuals with chronic pain, but the importance of psychosocial resources for the development of pain is sparsely examined, especially sex and gender differences. The aim of this study was to investigate associations between psychosocial resources and sex on the development of frequent pain in a general population sample, and to deepen the knowledge about sex and gender patterns.

### Methods

A sample from the Swedish Health Assets Project, a longitudinal cohort study, included self-reported data from 2263 participants, 53% women, with no frequent pain at baseline. The outcome variable was frequent pain at 18–months follow-up. Psychosocial resources studied were general self-efficacy, instrumental and emotional social support. Log binomial regressions in a generalised linear model were used to calculate risk ratios (RRs), comparing all combinations of men with high psychosocial resources, men with low psychosocial resources, women with high psychosocial resources and women with low psychosocial resources.

### Results

Women with low psychosocial resources had higher risk of frequent pain at follow-up compared to men with high resources: general self-efficacy RR 1.82, instrumental social support RR 2.33 and emotional social support RR 1.94. Instrumental social support was the most important protective resource for women, emotional social support was the most important one for men. Results were discussed in terms of gender norms.

**Data Availability Statement:** The data underlying this study cannot be made freely available as they are subject to secrecy in accordance with the

Swedish Public Access to Information and Secrecy Act (chapter 24, § 3 and 8). The Health Assets Project database is stored at the Swedish National Data Service (https://snd.gu.se/en) under the registration number SND0870. Requests to make data available to reproduce the findings in the study should be made to snd@gu.se.

**Funding:** The author(s) received no specific funding for this work.

**Competing interests:** The authors have declared that no competing interests exist.

## Conclusions

The psychosocial resources general self-efficacy, instrumental and emotional support predicted the risk of developing frequent pain differently among and between men and women in a general population sample. The results showed the importance of studying sex and gender differences in psychological and not least social predictors for pain.

## Introduction

Chronic pain is prevalent in the general population and prevention of chronic pain development is a key task for health care and public health [1–3]. Over the last decades there has been a growing interest to elaborate on the biopsychosocial model of chronic pain but there is still a knowledge gap when it comes to psychosocial resources as predictors of chronic pain, and the role of sex and gender [4–7].

Studies have demonstrated sex differences in psychosocial resources, in the general population and in clinical pain populations. Generally, men have higher self-efficacy than women [8–10], and women give and receive more social support than men [7, 11, 12]. Self-efficacy is an individual's belief to be able to achieve goals and master stressful challenges [13]. Social support denotes assistance and support from others when needed, including emotional social support (e.g., someone listens, shows empathy) and instrumental social support (tangible and practical support) [14]. Gendered expectations are suggested as an explanation for observed sex differences in psychosocial resources [4]. The concept of gender, the "social sex", is based on socially constructed expectations about how individuals enact femininity and masculinity. These expectations affect how individuals behave and how they are treated, by society and health care [4, 15]. Gendered expectations affect pain perception, as well as pain coping [4, 16]. Consequently, gender itself can be understood as a social factor with significance for pain and gendered expectations may impact on other psychosocial aspects of pain as well.

For instance, self-efficacy is, like self-confidence, initiative and decisiveness associated with traditional masculinity [4, 17], which may explain why men generally show higher self-efficacy than women [8, 9]. Social support can include emotional social support (to listen, to care) and instrumental (tangible, practical) social support [11]. Emotional social support is often associated with traditional femininity, both as an expression for emotionality and in relation to caring for and looking after others [4, 11]. However, a gendered denotation is not as obvious when it comes to instrumental social support. Social support per se may express female gender expectations, but instrumentality, on the other hand, is related to traditional masculinity like agency, control and "getting things done" [4, 16]. In pain research, instrumental and emotional social support are often analysed as one concept.

Different coping strategies for men and women with pain have been described, with men using more distractive behaviours and problem-focused strategies, while women use more emotion-focused strategies and social support [6]. However, it is not fully understood if psychosocial factors protect from chronic pain or increase the development of chronic pain likewise for men and women, and if potential sex patterns in these processes can be explained by gendered expectations [4, 5]. A better understanding of how psychosocial factors either protect from or increase the risk of developing frequent pain among men and women will contribute to enhanced pain prevention and treatment.

The overarching aim of this study was to deepen the knowledge about sex and gender patterns in the associations between pain and the psychosocial factors general self-efficacy (GSE), instrumental social support (ISS) and emotional social support (ESS). More specifically, to

investigate the associations between sex and psychosocial factors on the development of frequent pain in a general population sample.

## Methods

For this longitudinal study we used data from the "Health Assets Project" (HAP), described in detail by Holmgren et al. in 2010 [18]. HAP is a longitudinal cohort study with data collected twice, in February–April 2008 and September–November 2009, in Västra Götaland, Sweden. The study was conducted in accordance with the principles stated in the Declaration of Helsinki, with ethical approval received by the Regional Ethical Review Board of the University of Gothenburg in Sweden (registration number 039–08) and informed consent obtained from all participants.

### Participants

At baseline, questionnaires were sent to a random general population sample (n = 7984), with a response rate of 50.4%. The sample included in the current study consisted of those reporting no frequent pain at baseline and valid answers on pain at follow-up (n = 2263, 53% women) (Fig 1).

### Outcome

*Frequent pain* in the back or neck/shoulder at follow-up was measured by the following question: "How often have you had the following symptoms during the past twelve month?" Two

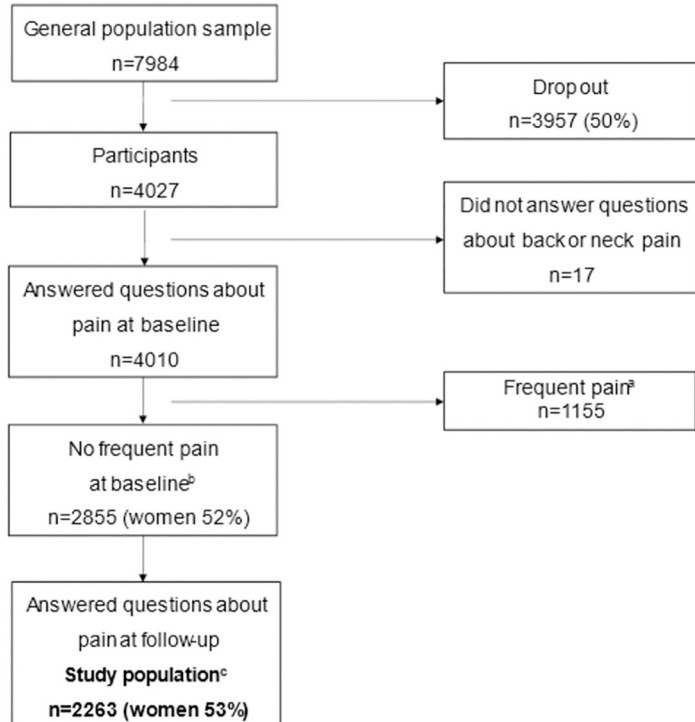

**Fig 1. Flowchart showing the inclusion of the study population, a general population sample from the Swedish Health Assets Project (HAP).** [a] pain nearly every day or now and again during the week, during the past twelve months, in the back or neck/shoulder. [b] pain now and again during the month or almost never or never, during the past twelve months, in the back or neck/shoulder. [c] individuals with no frequent pain at baseline who rated their pain in the back or neck/shoulder at follow-up.

of the listed options were "back pain, sciatica" and "neck and/or shoulder pain". Possible answers were "nearly every day", "now and again during the week", "now and again during the month" or "almost never or never". If the answer was "nearly every day" or "now and again during the week" for back pain or neck/shoulder pain, it was coded as frequent pain. Pain was used as a binary variable, including frequent pain and no frequent pain.

## Predictors

*General self-efficacy (GSE)* was measured with the General Self-Efficacy Scale (GSE scale) [19]. The Swedish version of the GSE scale, which has been validated in 2012 was used [9]. Mean scores were dichotomized into a 25/75% distribution. The cut-off was set at 2.70. Scores ranging from 1.00–2.70 were coded as low GSE, scores ranging from 2.71–4.00 were coded as high GSE [20]. High and low GSE were combined with sex to construct a predictor with four categories: Men with high GSE, men with low GSE, women with high GSE, women with low GSE.

*Social support* was measured with four questions based on the ENRICHD Social Support Inventory (ESSI), a valid and reliable screening tool [21]. The answers to the questions "Do you have a relative or a friend who is willing to help you if you are sick?" and "Do you have a friend or relative who is willing to help you if you need to borrow 15 000 SEK?" were combined to instrumental social support (ISS). The answers to the questions "Do you have a friend or relative who is willing to help you if you want company?" and "Do you have a friend or relative who is willing to help you if you want to talk with someone about personal problems?" were combined to emotional social support (ESS).

The questions could be answered with yes, no or do not know. The answers were dichotomized for ISS and ESS respectively. When a participant answered the two included questions with "yes + yes" or "yes + do not know" it was coded as strong ISS/ESS respectively. The combinations "no + no", "no + do not know", "no + yes", "do not know + do not know" were coded as weak ISS/ESS respectively. Strong and weak ISS were combined with sex to construct a predictor with four categories: Men with strong ISS, men with weak ISS, women with strong ISS, women with weak ISS. The same model was applied to ESS.

In summary, three different predictors were used: GSE, ISS and ESS. Each predictor included four values.

Age (3 categories: 19–30 years, 31–50 years, 51–63 years), level of education (3 categories: university/higher education, upper secondary school, compulsory school) and place of birth (2 categories: Nordic countries, other countries) were used as covariates.

## Statistical analysis

SPSS Statistics (version 28) were used for descriptive statistics and log binomial linear regressions. The outcome variable was frequent pain at follow-up. The psychosocial variables GSE, ISS and ESS at baseline were used as predicting variables. Each of the predicting variables included four possible categories: Men with high/strong GSE/ISS/ESS, men with low/weak GSE/ISS/ESS, women with high/strong GSE/ISS/ESS and women with low/weak GSE/ISS/ESS.

Log binomial regression was used in a generalised linear model estimating the regression parameters (unstandardised regression weight), their 95% confidence intervals (CIs) and p-values, for the associations between psychosocial factors at baseline and frequent pain at follow-up. Results were presented unadjusted and adjusted for the covariates age, level of education and place of birth. Significance levels were set at $p < 0.05$.

Using estimated model means, generated in the log binomial regression model, risks and their 95% CIs were calculated for each value, and RRs for all pairs of values within each

variable: Women with low GSE compared to women with high GSE, women with low GSE compared to men with low GSE, women with low GSE compared to men with high GSE, women with high GSE compared to men with low GSE, women with high GSE compared to men with high GSE, men with low GSE compared to men with high GSE. The same principle applied to ISS and ESS.

RRs were used since they are intuitive in interpretation, will not over-estimate effect sizes as odds ratios can do and therefore have been recommended for prospective designs when the outcome variable is common (> 10%) [22–24].

## Results

Women were higher educated than men, with 44% at university level, compared with 37% for men. More women than men (15% versus 10%) reported frequent pain at follow-up. In addition, more women than men reported low GSE, strong ISS and strong ESS (Table 1).

Men with high psychosocial resources (GSE, ISS and ESS) at baseline had the lowest risk of frequent pain at follow-up and women with low resources had the highest risk. However, the risks associated with the three predictors differed between and within the groups of men and

**Table 1. Characteristics of the study population.**

|  | Total | | Men | | Women | |
|---|---|---|---|---|---|---|
|  | **n** | **(%)** | **n** | **(%)** | **n** | **(%)** |
| **Total** | 2263 | (100) | 1069 | (47) | 1194 | (53) |
| **Pain at follow-up (2009)** |  |  |  |  |  |  |
| Frequent pain | 283 | (13) | 109 | (10) | 174 | (15) |
| No frequent pain | 1980 | (88) | 960 | (90) | 1020 | (85) |
| **Age** |  |  |  |  |  |  |
| 19–30 years | 438 | (19) | 196 | (18) | 242 | (20) |
| 31–50 years | 1025 | (45) | 488 | (46) | 537 | (45) |
| 51–63 years | 800 | (35) | 385 | (36) | 415 | (35) |
| **Education** |  |  |  |  |  |  |
| University/higher education | 914 | (40) | 391 | (37) | 523 | (44) |
| Upper secondary school | 960 | (42) | 488 | (46) | 472 | (40) |
| Compulsory school | 370 | (16) | 180 | (17) | 190 | (16) |
| [Missing] | 19 | (1) | 10 | (1) | 9 | (1) |
| **Place of birth** |  |  |  |  |  |  |
| Nordic countries | 2094 | (93) | 984 | (92) | 1110 | (93) |
| Other countries | 169 | (8) | 85 | (8) | 84 | (7) |
| **General self-efficacy (GSE)** |  |  |  |  |  |  |
| Low GSE | 450 | (20) | 177 | (17) | 273 | (23) |
| High GSE | 1782 | (79) | 880 | (82) | 902 | (76) |
| [Missing] | 31 | (1) | 12 | (1) | 19 | (2) |
| **Instrumental social support (ISS)** |  |  |  |  |  |  |
| Weak ISS | 328 | (15) | 181 | (17) | 147 | (12) |
| Strong ISS | 1896 | (84) | 867 | (81) | 1029 | (86) |
| [Missing] | 39 | (2) | 21 | (2) | 18 | (2) |
| **Emotional social support (ESS)** |  |  |  |  |  |  |
| Weak ESS | 199 | (9) | 119 | (11) | 80 | (7) |
| Strong ESS | 2035 | (90) | 931 | (87) | 1104 | (93) |
| [Missing] | 29 | (1) | 19 | (2) | 10 | (1) |

**Table 2. The associations between general self-efficacy (GSE), instrumental social support (ISS) and emotional social support (ESS) at baseline and risk of frequent pain at follow-up.**

| | Unadjusted model | | | Adjusted model (age, place of birth, education) | | |
|---|---|---|---|---|---|---|
| | B | 95% CI | p-value | B | 95% CI | p-value |
| **GSE** | | | | | | |
| Men high GSE = ref | | | | | | |
| Men low GSE | 0.17 | -0.31; 0.60 | 0.454 | 0.19 | -0.29; 0.61 | 0.417 |
| Women high GSE | 0.29 | 0.03; 0.55 | 0.031 | 0.31 | 0.06; 0.58 | 0.018 |
| Women low GSE | 0.59 | 0.26; 0.90 | <0.001 | 0.60 | 0.27; 0.92 | <0.001 |
| Intercept | -2.30 | -2.51; -2.11 | <0.001 | -2.02 | -2.51; -1.57 | <0.001 |
| **ISS** | | | | | | |
| Men strong ISS = ref | | | | | | |
| Men weak ISS | 0.38 | -0.06; 0.78 | 0.076 | 0.30 | -0.15; 0.71 | 0.168 |
| Women strong ISS | 0.36 | 0.10; 0.62 | 0.007 | 0.37 | 0.11; 0.63 | 0.005 |
| Women weak ISS | 0.86 | 0.49; 1.22 | <0.001 | 0.85 | 0.47; 1.20 | <0.001 |
| Intercept | -2.36 | -2.57; -2.16 | <0.001 | -2.08 | -2.57; -1.62 | <0.001 |
| **ESS** | | | | | | |
| Men strong ESS = ref | | | | | | |
| Men weak ESS | 0.76 | 0.32; 1.16 | <0.001 | 0.69 | 0.23; 1.10 | 0.002 |
| Women strong ESS | 0.47 | 0.23; 0.73 | <0.001 | 0.49 | 0.24; 0.74 | <0.001 |
| Women weak ESS | 0.66 | 0.10; 1.14 | 0.012 | 0.66 | 0.09; 1.14 | 0.012 |
| Intercept | -2.41 | -2.62; -2.21 | <0.001 | -2.12 | -2.61; -1.67 | <0.001 |

Log binomial regression was used to model the risk of frequent pain at follow-up in a random general population sample with no frequent pain at baseline. Predictors in the model, each combined with sex: General self-efficacy (GSE), instrumental social support (ISS), emotional social support (ESS).

women. Results are summarised in Table 2. The adjustment for age, place of birth and educational level altered the results only marginally and all results presented in subsequent figures and tables are based on adjusted analyses.

## General self-efficacy

The risk of frequent pain ranged from 0.21 among women with low GSE to 0.11 among men with high GSE (Fig 2). RRs showed that women with low GSE had an 82% higher risk of frequent pain at follow-up compared to men with high GSE, CIs were contiguous but not overlapping (Fig 2, Table 3). Women with high GSE had a 37% higher risk than men with high GSE, but the CIs were overlapping. The difference in the risk of frequent pain at follow-up between men with low and men with high GSE was not statistically significant (Table 2), and the CIs were greatly overlapping (Fig 2, Table 3).

## Instrumental social support

The risk of frequent pain ranged from 0.26 among women with weak ISS to 0.11 among men with strong ISS (Fig 3). RRs showed that women with weak ISS had a 133% higher risk of frequent pain at follow-up compared to men with strong ISS, CIs not overlapping (Fig 3, Table 4). Women with weak ISS had also a 62% higher risk than women with strong ISS, CIs marginally overlapping. Women with strong ISS had 44% higher risk than men with strong ISS, CIs marginally overlapping. The difference in the risk of frequent pain at follow-up between men with weak and men with strong ISS was not statistically significant (Table 2), and the CIs were greatly overlapping (Fig 3, Table 4).

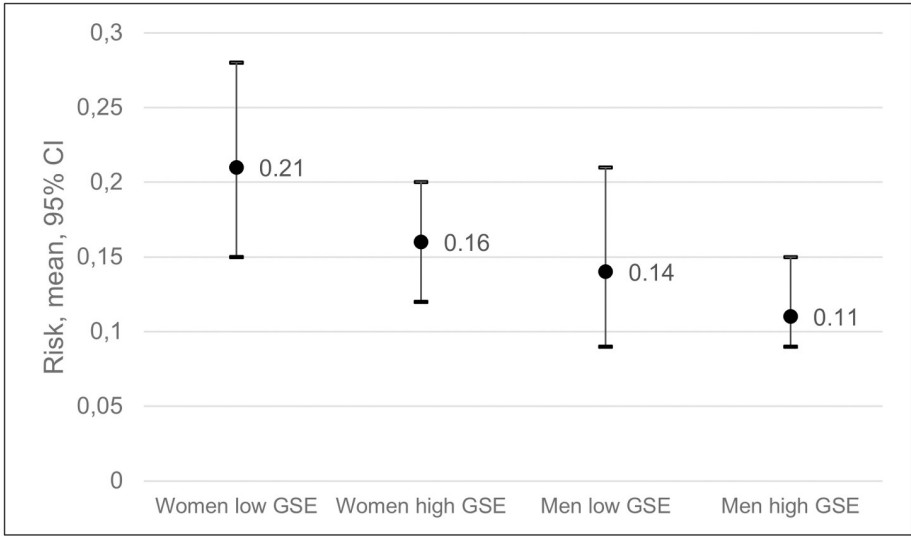

**Fig 2. Risk of frequent pain at follow-up among women with low general self-efficacy (GSE), women with high GSE, men with low GSE and men with high GSE at baseline.**

## Emotional social support

The risk of frequent pain ranged from 0.21 among men with weak ESS to 0.1 among men with strong ESS (Fig 4). RRs showed that men with weak ESS had a 100% higher risk of having frequent pain at follow-up compared to men with strong ESS, CIs not overlapping (Fig 4, Table 5). Women with strong ESS had a 63% higher risk than men with strong ESS, CIs not overlapping, and women with weak ESS had a 94% higher risk, CIs marginally overlapping.

## Discussion

This study aimed to deepen the knowledge about sex and gender patterns in the psychosocial factors general self-efficacy (GSE), instrumental social support (ISS) and emotional social support (ESS) in relation to development of frequent pain at follow-up in a general population sample with no frequent pain at baseline.

We found that women with low psychosocial resources had considerably higher risk of frequent pain at follow-up compared to men with high resources. The risk of frequent pain was also higher among women with high resources, compared to men with high resources. Generally, more women than men develop chronic pain [6], but our results showed that the

**Table 3. Risk ratios (RRs) and confidence intervals (CIs) for frequent pain at follow-up, associated with baseline general self-efficacy (GSE) among all possible combinations of GSE and sex.**

| | Risk ratios (RRs) | Confidence intervals (CIs) |
|---|---|---|
| Women low GSE/women high GSE | 1.33 | 0.15; 0.28 / 0.12; 0.20 |
| Women low GSE/men low GSE | 1.51 | 0.15; 0.28 / 0.09; 0.21 |
| Women low GSE/men high GSE | **1.82** | **0.15; 0.28 / 0.09; 0.15[a]** |
| Women high GSE/men low GSE | 1.14 | 0.12; 0.20 / 0.09; 0.21 |
| Women high GSE/men high GSE | 1.37 | 0.12; 0.20 / 0.09; 0.15 |
| Men low GSE/men high GSE | 1.20 | 0.09; 0.21 / 0.09; 0.15 |

[a] CIs contiguous

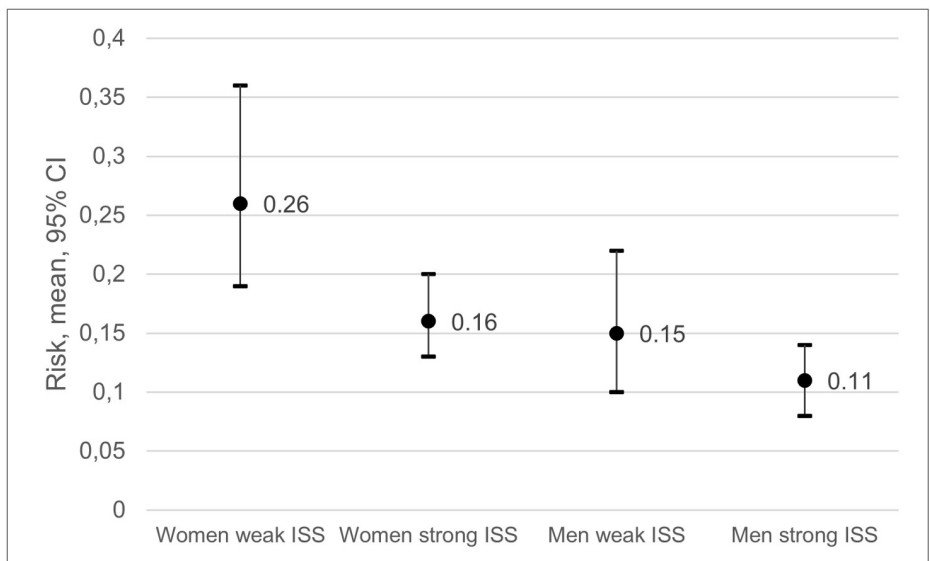

**Fig 3. Risk of frequent pain at follow-up among women with weak instrumental social support (ISS), women with strong ISS, men with weak ISS and men with strong ISS at baseline.**

combination of low psychosocial resources and being a woman increased the risk of frequent pain substantially. Hence, women with low psychosocial resources may be particularly vulnerable for developing frequent pain.

## Sex and gender patterns for men

There were no statistically significant differences in risks between men with high and men with low resources in GSE and ISS. GSE has been described as a resource for individuals with chronic pain, beneficial for their coping with stress and pain [25–27]. However, even if studies found that GSE in patients with chronic pain predicts pain behaviour, perceived pain and disability [25, 28], this study showed that for men, in a general population sample, low GSE or weak ISS did not seem to play a major role for the development of frequent pain. Self-efficacy and instrumentality are traditionally associated with masculinity [4]. To have strong resources in characteristics that are associated with traditional masculinity did not provide men with an extra protection against frequent pain.

**Table 4. Risk ratios (RRs) and confidence intervals (CIs) for frequent pain at follow-up, associated with baseline instrumental social support (ISS) among all possible combinations of ISS and sex.**

|  | Risk ratios (RRs) | Confidence intervals (CIs) |
|---|---|---|
| Women weak ISS/women strong ISS | **1.62** | **0.19; 0.36 / 0.13; 0.20**[a] |
| Women weak ISS/men weak ISS | 1.73 | 0.19; 0.36 / 0.10; 0.22 |
| Women weak ISS/men strong ISS | **2.33** | **0.19; 0.36 / 0.08; 0.14**[b] |
| Women strong ISS/men weak ISS | 1.07 | 0.13; 0.20 / 0.10; 0.22 |
| Women strong ISS/men strong ISS | **1.44** | **0.13; 0.20 / 0.08; 0.14**[a] |
| Men weak ISS/men strong ISS | 1.35 | 0.10; 0.22 / 0.08; 0.14 |

[a] CIs marginally overlapping

[b] CIs not overlapping

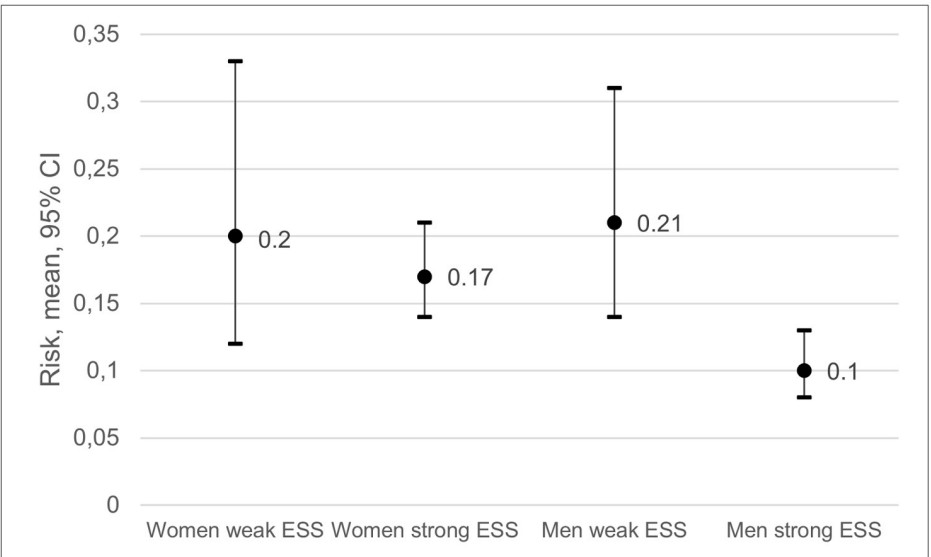

**Fig 4. Risk of frequent pain at follow-up among women with weak emotional social support (ESS), women with strong ESS, men with weak ESS and men with strong ESS at baseline.**

In contrast, the patterns for ESS were different. Men with weak ESS had a substantially higher risk of developing frequent pain compared to men with strong ESS. This was a surprising result. Emotionality and ESS are strongly associated with traditional femininity [4]. Feminine values (as emotionality and sensitivity) are often underestimated, and masculine values (as determination and action) generally have a higher status in society and health care [29]. Men seem to receive most of their emotional support from women, in partner relations [30]. This support might easily be taken for granted and men might be unconscious about its significance (for pain). Greater attention to the importance of ESS for men could have two functions: 1) to strengthen men's protective psychosocial resources and 2) to challenge traditional gender norms promoting traditional masculinity as most desirable or the most beneficial attitude for men.

## Sex and gender patterns for women

Among women, weak ISS was associated with the highest risk of developing frequent pain at follow-up. Women with weak ISS had a considerably higher risk of developing frequent pain

**Table 5. Risk ratios (RRs) and confidence intervals (CIs) for frequent pain at follow-up, associated with baseline emotional social support (ESS) among all possible combinations of ESS and sex.**

|  | Risk ratios (RRs) | Confidence intervals (CIs) |
|---|---|---|
| Women weak ESS/women strong ESS | 1.19 | 0.12; 0.33 / 0.14; 0.21 |
| Women weak ESS/men weak ESS | 0.97 | 0.12; 0.33 / 0.14; 0.31 |
| Women weak ESS/men strong ESS | **1.94** | **0.12; 0.33 / 0.08; 0.13**[a] |
| Women strong ESS/men weak ESS | 0.81 | 0.14; 0.21 / 0.14; 0.31 |
| Women strong ESS/men strong ESS | **1.63** | **0.14; 0.21 / 0.08; 0.13**[b] |
| Men weak ESS/men strong ESS | **2.00** | **0.14; 0.31 / 0.08; 0.13**[b] |

[a] CIs marginally overlapping

[b] CIs not overlapping

than women with strong ISS. In contrast, women with weak ESS had only a slightly higher risk of frequent pain at follow-up. This is an unexpected result. Women generally give and receive more social support than men and women with chronic pain use social support more often than men as a coping strategy [6, 7, 12, 31]. The "stress-buffering model" [1, 11, 32] states that social support can lower stress, which in turn can have a beneficial effect on pain [1, 32] and quality of life [11]. Our study is, to the best of our knowledge, the first one showing that strong ISS may be more important than strong ESS for women as a protective factor regarding pain. ESS is associated with traditional femininity; ISS is more ambiguous than ESS in terms of gender coding. Maybe it is not as obvious for women who to address to when they need instrumental or tangible support. It has also been shown that particularly women face multiple expectations in everyday life to manage paid work, household duties, childcare, their body, their pain etc. [29, 33], which can require physical and mental strength. It is possible that instrumental support, relieving some of the concrete burdens, can provide a protection against the development of frequent pain, especially for women. Our results are not sufficiently far-reaching to support or dismiss these assumptions, and to further explore the meaning of different kinds of social support for women with and without frequent pain in future research is highly recommended.

## Gender-balanced social support

Results on what effect social support has on men´s and women's pain have been inconsistent in earlier research. Social support has been denoted as an asset for patients with pain [1, 34], and the lack of social support has been described as a risk factor for depression, especially for women [35]. Others have suggested that social support might be a maladaptive coping strategy for women with pain [7], might increase men's pain [36] or could be either beneficial or disadvantageous for men and women in different circumstances [1, 32]. Social support includes both ISS and ESS but in pain research they are frequently studied unseparated, e.g. [34]. In this study, men and women showed different patterns for ISS and ESS as predictors for frequent pain. These results indicate that ISS and ESS should be analysed separately, and that men and women might benefit from ISS and ESS differently. In addition, intersectional future research on different kinds of social support, combined with sex, also might explain some of the inconsistencies in earlier pain research.

As ESS is associated with femininity and ISS might be associated with masculinity, our results indicate that traditional feminine resources might lower the risk for frequent pain, especially for men, and traditional masculine resources might lower the risk for frequent pain, especially for women. Our results suggest that a gender-balanced access to psychosocial resources might be beneficial for men and women.

## Methodological considerations

The measurement of social support in pain research is fragmented, often addressed by very few questions. This study was no exception. As social support is associated with gendered expectations, it might be wise to reconsider the questions asked in social support scales. Nolan-Hoeksema (2012) argued that men might seek invisible emotional and practical support through, for instance, shared activities, to be able to maintain their sense of masculinity [37]. It might also be possible that women intuitively score higher on items associated with high traditional femininity and that men report the other way around, because of gendered expectations. However, in this study, men and women reported generally strong ISS and ESS, and this study showed, in particular, the need to further explore different aspects of social support, including sex and gender differences (and similarities) more detailed and in depth.

We examined the role of psychosocial resources for frequent/no frequent pain. Most pain research addresses chronic pain. Our definition of frequent pain is close to the definition of chronic pain [38] but not identical. Consequently, our results may not be generalised to populations with or without chronic pain without caution but can generate hypotheses that should be further explored in (non)chronic pain populations.

We adjusted for age, level of education and place of birth, none of them altered the results more than marginally. Other factors that we did not control for, like partner relations or household income might have influenced the results. Studies have also shown a strong association between depression, anxiety and chronic pain [1, 26, 39], as well as a prospective association between low mental well-being and the development of chronic pain [1, 40]. We could have adjusted for mental well-being but chose not to, as we expected mental well-being to function as a mediator.

Among the strengths of this study is its longitudinal prospective design, with 2263 participants from the general population included, generating new insights in the important field of pain prevention. This study analysed sex differences and discussed gender aspects as possible explanations for sex differences. This is one of different possible ways of broadening the knowledge about men and women with pain.

## Conclusions

Analyses of general self-efficacy, instrumental and emotional social support, combined with sex, provided a nuanced picture of men's and women's risk of frequent pain. Being a woman and having low psychosocial resources indicated the highest risk.

Strong emotional social support was the most protective resource for men. For women, it was strong instrumental social support. These results suggest that a gender-balanced access to psychosocial resources might be beneficial for men and women. It is reasonable to expect that both men and women, but especially women with low resources could benefit from initiatives aimed to strengthen their self-efficacy, instrumental and emotional social support. There is also a need to further explore not only psychological but also social predictors of pain, among men and women.

## Author Contributions

**Conceptualization:** Anke Samulowitz, Inger Haukenes, Anna Grimby-Ekman, Stefan Bergman, Gunnel Hensing.

**Formal analysis:** Anke Samulowitz.

**Methodology:** Anke Samulowitz, Inger Haukenes, Anna Grimby-Ekman, Stefan Bergman, Gunnel Hensing.

**Writing – original draft:** Anke Samulowitz.

**Writing – review & editing:** Anke Samulowitz, Inger Haukenes, Anna Grimby-Ekman, Stefan Bergman, Gunnel Hensing.

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
