## [Decision Letter · Decision Letter 0]

28 Nov 2022

PONE-D-22-26227Psychosocial resources predict frequent pain differently for men and women: a prospective cohort studyPLOS ONE

Dear Dr. Samulowitz,

Thank you for submitting your manuscript to PLOS ONE. After careful consideration, we feel that it has merit but does not fully meet PLOS ONE’s publication criteria as it currently stands. Therefore, we invite you to submit a revised version of the manuscript that addresses the points raised during the review process.

We look forward to receiving your revised manuscript.

Kind regards,

Hadi Ghasemi

Academic Editor

PLOS ONE

Journal Requirements:

Reviewers' comments:

Reviewer's Responses to Questions

**Comments to the Author**

1. Is the manuscript technically sound, and do the data support the conclusions?

Reviewer #1: Yes

Reviewer #2: Yes

2. Has the statistical analysis been performed appropriately and rigorously? 

Reviewer #1: Yes

Reviewer #2: Yes

3. Have the authors made all data underlying the findings in their manuscript fully available?

Reviewer #1: No

Reviewer #2: Yes

4. Is the manuscript presented in an intelligible fashion and written in standard English?

Reviewer #1: Yes

Reviewer #2: Yes

5. Review Comments to the Author

Reviewer #1: This is an interesting paper on psychosocial resources predicting men's and women’s frequent pain differently. I liked reading the paper and learnt from it. Some of my comments are:

• In the abstract, explain what you mean by psychosocial resources upfront so that readers can understand what they can expect.

• Please provide the citations following proper citation style- 1-3, 4-7, not the way citations should be provided.

• Define self-efficacy and social support in the introduction, paragraph 2, page 3, authors mention: “Generally, men have higher self-efficacy than women (8-10), and women give and receive more social support than men (7, 11, 12).”

Also, define what you mean by instrumental social support upfront so that readers can follow smoothly.

• Explain the Swedish version, validated in 2012 measurement, very clearly so readers can understand better.

• In this paragraph containing “The scores of two of the questions (“Do you have a relative or a friend who is willing to help you if you are sick?”: I think it would be better not to use the word “scores” as it confuses readers with what “scores” the authors are talking about. Instead, authors can say “answers” and later explain how the answers are combined to get a 0-1 score.

• Explain what the categories are. - Age (3 categories), education (3 categories), and country of birth (2 categories) were used as control variables.

• It will be easier to understand the dependent and the independent variables if the authors present an equation. I understood frequent pain = f(baseline score of psychosocial resources by sex, control). Is frequent pain a binary measurement? How are all the questions for measuring dependent variables combined?

• Are these psychosocial resources run separately during the estimation? I thought the model would likely be frequent pain = f(men and women GSE, ISS, ESS, control). The separating solid black lines in Table 2 between GSE, ISS, and ESS are confusing.

• Overall comments: check the citation/grammar.

Reviewer #2: The article is interesting i would suggest to, if possible add the time ( duration ) should be taken into account to something like pain.

The subject is current and the way you organized the article is quite well structured.

6. PLOS authors have the option to publish the peer review history of their article (what does this mean?). If published, this will include your full peer review and any attached files.

Reviewer #1: No

Reviewer #2: **Yes: **Andre Mariz de Almeida

---

## [Author Response · Author response to Decision Letter 0]

11 Dec 2022

Many thanks for your valuable considerations. They were most helpful to improve the manuscript. Below, we have addressed your comments, suggestions and questions. 

Academic editor 

 We have adjusted the manuscript according to PLOS ONE’s style requirements

In your Data Availability statement, you have not specified where the minimal data set underlying the results described in your manuscript can be found. Important: If there are ethical or legal restrictions to sharing your data publicly, please explain these restrictions in detail.

 The data underlying this study cannot be made freely available as they are subject to secrecy in accordance with the Swedish Public Access to Information and Secrecy Act (chapter 24, § 3 and 8). 

The Health Assets Project database is stored at the Swedish National Data Service (https://snd.gu.se/en) under the registration number SND0870. Requests to make data available to reproduce the findings in the study should be made to snd@gu.se. 

Reviewers' comments: 

1. Is the manuscript technically sound, and do the data support the conclusions?

Reviewer #1: Yes

Reviewer #2: Yes 

2. Has the statistical analysis been performed appropriately and rigorously?

Reviewer #1: Yes

Reviewer #2: Yes 

3. Have the authors made all data underlying the findings in their manuscript fully available?

Reviewer #1: No

Reviewer #2: Yes 

 The data underlying this study cannot be made freely available as they are subject to secrecy in accordance with the Swedish Public Access to Information and Secrecy Act (chapter 24, § 3 and 8). The Health Assets Project database is stored at the Swedish National Data Service (https://snd.gu.se/en) under the registration number SND0870. Requests to make data available to reproduce the findings in the study should be made to snd@gu.se. 

4. Is the manuscript presented in an intelligible fashion and written in standard English?

Reviewer #1: Yes

Reviewer #2: Yes 

5. Review Comments to the Author 

In the abstract, explain what you mean by psychosocial resources upfront so that readers can understand what they can expect.

 We have added “Psychosocial resources, psychological and social factors like self-efficacy and social support” in the abstract

Please provide the citations following proper citation style- 1-3, 4-7, not the way citations should be provided.

 We have now uploaded the PLOS citation template in endnotes and used it throughout the manuscript

Define self-efficacy and social support in the introduction, paragraph 2, page 3, authors mention: “Generally, men have higher self-efficacy than women (8-10), and women give and receive more social support than men (7, 11, 12).” Also, define what you mean by instrumental social support upfront so that readers can follow smoothly.

 Thank you for your important consideration. We agree that definitions were missing and have added definitions for general self-efficacy, social support, emotional and instrumental social support

Explain the Swedish version, validated in 2012 measurement, very clearly so readers can understand better.

 We have rephrased the sentence

In this paragraph containing “The scores of two of the questions (“Do you have a relative or a friend who is willing to help you if you are sick?”: I think it would be better not to use the word “scores” as it confuses readers with what “scores” the authors are talking about. Instead, authors can say “answers” and later explain how the answers are combined to get a 0-1 score.

 Thank you for your suggestion. We agree that it makes the text more comprehensible. We have changed “scores” to “answers”

Explain what the categories are. - Age (3 categories), education (3 categories), and country of birth (2 categories) were used as control variables.

 We have added information about the categories

It will be easier to understand the dependent and the independent variables if the authors present an equation. I understood frequent pain = f(baseline score of psychosocial resources by sex, control). Is frequent pain a binary measurement? How are all the questions for measuring dependent variables combined?

 Yes, frequent pain is a binary variable. We agree that there is a need of clarification and have added information about how the questions for measuring the dependent variables are combined

Are these psychosocial resources run separately during the estimation? I thought the model would likely be frequent pain = f(men and women GSE, ISS, ESS, control). The separating solid black lines in Table 2 between GSE, ISS, and ESS are confusing.

 The analyses for GSE, ISS and ESS were run separately. As we conducted three different estimations, those were presented as three different parts in Table 2 

Overall comments: check the citation/grammar.

 We have checked citations and grammar carefully to ensure correct language

Reviewer 2 

The article is interesting i would suggest to, if possible add the time ( duration ) should be taken into account to something like pain. The subject is current and the way you organized the article is quite well structured.

 We agree that duration is an important aspect in pain. The time aspects available in this study were questions about pain frequency during the past twelve month. We agree, in future studies, pain duration and the associations between pain duration, self-efficacy and social support should be explored in more detail

6. PLOS authors have the option to publish the peer review history of their article (what does this mean?). If published, this will include your full peer review and any attached files.

If you choose “no”, your identity will remain anonymous, but your review may still be made public. 

 Thank you for this information

---

## [Editor Report · Decision Letter 1]

6 Mar 2023

Psychosocial resources predict frequent pain differently for men and women: a prospective cohort study

PONE-D-22-26227R1

Dear Dr. Anke Samulowitz,

We’re pleased to inform you that your manuscript has been judged scientifically suitable for publication and will be formally accepted for publication once it meets all outstanding technical requirements.

Kind regards,

Hadi Ghasemi

Academic Editor

PLOS ONE
---

## [Editor Report · Acceptance letter]

9 Mar 2023

PONE-D-22-26227R1 

Psychosocial resources predict frequent pain differently for men and women: a prospective cohort study 

Dear Dr. Samulowitz:

I'm pleased to inform you that your manuscript has been deemed suitable for publication in PLOS ONE. Congratulations! Your manuscript is now with our production department. 

Kind regards, 

on behalf of

Dr. Hadi Ghasemi 

Academic Editor

PLOS ONE